# Activation of σ1-Receptors by R-Ketamine May Enhance the Antidepressant Effect of S-Ketamine

**DOI:** 10.3390/biomedicines11102664

**Published:** 2023-09-28

**Authors:** Hans O. Kalkman

**Affiliations:** Retired Pharmacologist, Gänsbühlgartenweg 7, 4132 Muttenz, Switzerland; hans.kalkman@bluewin.ch

**Keywords:** NMDA channel blockade, opioid μ-receptors, σ1 agonist, BDNF, microglia, AGC kinase

## Abstract

Ketamine is a racemic mixture composed of two enantiomers, S-ketamine and R-ketamine. In preclinical studies, both enantiomers have exhibited antidepressant effects, but these effects are attributed to distinct pharmacological activities. The S-enantiomer acts as an NMDA-channel blocker and as an opioid μ-receptor agonist, whereas the R-enantiomer binds to σ1-receptors and is believed to act as an agonist. As racemate, ketamine potentially triggers four biochemical pathways involving the AGC-kinases, PKA, Akt (PKB), PKC and RSK that ultimately lead to inhibitory phosphorylation of GSK3β in microglia. In patients with major depressive disorder, S-ketamine administered as a nasal spray has shown clear antidepressant activity. However, when compared to intravenously infused racemic ketamine, the response rate, duration of action and anti-suicidal activity of S-ketamine appear to be less pronounced. The σ1-protein interacts with μ-opioid and TrkB-receptors, whereas in preclinical experiments σ1-agonists reduce μ-receptor desensitization and improve TrkB signal transduction. TrkB activation occurs as a response to NMDA blockade. So, the σ1-activity of R-ketamine may not only enhance two pathways via which S-ketamine produces an antidepressant response, but it furthermore provides an antidepressant activity in its own right. These two factors could explain the apparently superior antidepressant effect observed with racemic ketamine compared to S-ketamine alone.

## 1. Introduction

Numerous studies have consistently supported the initial findings by Berman et al. [1] regarding the rapid onset of antidepressant activity displayed by intravenous ketamine [2,3]. Further studies have noted that the response is sustained, that the remission rate is superior to that of traditional antidepressants and that the compound is effective in treatment-resistant patients [2,3]. The publication by Berman and colleagues has sparked a search for the mechanism underlying the rapid and sustained antidepressant effects. Ketamine is not a single compound, but consists of a 50/50 mixture of R-ketamine and S-ketamine. Radioligand binding studies show that racemic ketamine binds to the phencyclidine site of the glutamate NMDA (N-methyl-D-aspartate) channel, to opioid μ-receptors and a chaperone protein known as the σ1-site [4]. Further investigations have revealed that S-ketamine blocks the NMDA channel and stimulates opioid receptors, whereas R-ketamine primarily binds to the σ1-site [5]. Although both enantiomers of ketamine lack affinity to glutamate AMPA receptors [5], these receptors play a role in the antidepressant responses to S- and R-ketamine (see sections below). In the clinic, S-ketamine administered as a nasal spray has demonstrated definite antidepressant activity [6]. However, the clinical impression is that intravenously infused racemic ketamine displays a superior response rate [7] and duration of action [8] compared to intranasal S-ketamine. It should be noted though that a true comparison would require a head-to-head study, which as of today has not been performed. Nevertheless, there may be qualitative differences, too, as racemic ketamine, unlike S-ketamine, has shown anti-suicidal activity [9]. The plasma levels of S-ketamine following nasal S-ketamine or intravenous racemic ketamine application cannot readily account for the qualitative and quantitative differences observed in treatment response [10]. Thus, alongside the ongoing debate regarding the mechanism of action of S-ketamine (whether NMDA channel blockade or μ-opioid receptor activity is responsible for its antidepressant effects) [11,12], the contribution of R-ketamine to the effects observed after racemic ketamine remains an open question. After analysis of the pharmacological data accumulated for the individual enantiomers, it seems likely that R-ketamine, due to its σ1-activity provides not only an additional antidepressant mechanism, but potentially enhances both the μ-opioid receptor activity and the brain-derived neurotrophic factor (BDNF) signaling process induced by S-ketamine. These two factors could explain the superior antidepressant effect observed with racemic ketamine compared to S-ketamine.

## 2. NMDA Blockade: Does It Induce Antidepressant Activity?

In preclinical studies, ketamine and NMDA channel blockers like phencyclidine (PCP), dizocilpine (MK-801), memantine or lanicemine were shown to enhance extracellular glutamate levels in the brain via a reduction in the suppressive effects of GABA-interneurons, which ultimately led to the formulation of “ketamine disinhibition hypothesis” (recently reviewed in [13,14]). It posits that ketamine’s pharmacological action involves stimulation of glutamatergic AMPA (α-amino-3-hydroxy-5-methylisoxazole-4-propionic acid) receptors, triggering the release of BDNF and correcting the suppressed synapse and neurite formation in the brain of depression patients [14,15]. The issue with this hypothesis is that it fails to explain why phencyclidine, dizocilpine, memantine or lanicemine, despite causing glutamatergic disinhibition, do not share the robust antidepressant activity of ketamine [15,16,17,18]. A partial explanation for the discrepancy may lie in the dose-response characteristics. Animal models have shown a narrow dose-response relationship for the antidepressant activity of ketamine, with the effect disappearing at moderately higher doses (described as “inverted U-shaped”) [19]. Several allosteric activators of AMPA channels, developed on the basis of the disinhibition hypothesis, also display a narrow inverted U-shaped dose-response relationship [20] and similarly failed to demonstrate antidepressant activity in clinical trials [21]. These findings indicate that agonist dosing at the AMPA channel is highly delicate, and while a rapid antidepressant-like effect may be observed in preclinical experiments with often just a single dose, the effect may not be reproducible in clinical settings. 

An alternative explanation for the antidepressant effect of S-ketamine involves the activation of μ-opioid receptors [4,5]. In preclinical experiments, subanesthetic doses of S-ketamine, but not R-ketamine, induced locomotor activity (in an opioid receptor-dependent manner), psychomotor sensitization, conditioned place preference, and selectively increased metabolic activity and dopamine tone in the mPFC [5]. These findings strongly suggest that subanesthetic doses of S-ketamine activate μ-opioid receptors. Moreover, rats self-administered S-ketamine, but not R-ketamine, indicating a rewarding property of S-ketamine that most likely is due to μ-opioid receptor activation [5]. S-ketamine is rapidly converted to S-norketamine [22,23], so a possible μ-opioid receptor stimulation by this metabolite should be kept in mind, too. Although the potency and efficacy of S-norketamine at μ-opioid receptors remain to be determined, it is worth noting that the antidepressant effect of S-norketamine, unlike S-ketamine, was unaffected by AMPA blockade, which aligns with μ-opioid receptor activation but contradicts NMDA blockade as the mechanism of action [24]. Activation of μ-opioid receptors has been validated as a mechanism for antidepressant activity [25,26], as evidenced, for instance, by the registered antidepressant tianeptine [27,28]. Therefore, it is possible that S-ketamine exerts its antidepressant activity through two concurrent mechanisms: AMPA channel activation (indirectly resulting from NMDA blockade) and μ-opioid receptor activation. In clinical studies, the evidence is mixed: the opioid antagonist naltrexone blocked the ketamine-induced antidepressant effects in one study [29], but not in another [30]. 

## 3. R-Ketamine: Does It Contribute to the Clinical Profile of Ketamine? 

In preclinical studies, R-ketamine displayed higher potency and longer-lasting antidepressant-like action compared to S-ketamine [18,31,32,33], despite the lower affinity of R-ketamine for the NMDA-channel or the μ-opioid receptor [4,5]. This distinction also extends to anti-inflammatory activity, where R-ketamine is more active than S-ketamine (reviewed by [23]). In individuals with major depressive disorder (MDD) who respond to treatment, a single intravenous infusion of ketamine was shown to reduce serum IL-6 levels for up to 3 days [34]. Similarly, in patients receiving six ketamine infusions, the decrease in plasma levels of IL-17A and IL-6 correlated with symptom improvement [35]. These findings suggest that ketamine’s reduction of inflammation may contribute to its sustained antidepressant effects. Microglia cells represent the brain’s resident immune system [36,37,38]. Microglia cells are major players in the release of cytokines and the pruning of synapses [37,39] and are involved in the antidepressant activity of medications [40,41]. In mice that were susceptible to chronic social defeat stress, an inflammation-like impairment of CREB-mediated transcription of BDNF in microglia was observed [42]. R-ketamine activated the MEK-ERK kinase pathway and restored both microglial CREB activity and BDNF release [42]. Furthermore, BDNF, through stimulation of neuronal tropomyosin receptor kinase B (TrkB) receptors, restored the reduced dendritic spine density in mice susceptible to chronic social defeat stress [42]. Importantly, these effects were observed after a single intraperitoneal dose of R-ketamine, but not with S-ketamine [42]. Indeed, R-ketamine definitely increases TrkB signaling [43,44,45]. The normalization of spine density and synapse formation by R-ketamine is believed to underlie its long-lasting antidepressant-like effects observed in animal models [42].

The molecular target that is responsible for R-ketamine’s effects has not been definitively identified, but the available evidence suggests the σ1 receptor is a strong candidate. R-ketamine has a higher affinity for σ1 sites than S-ketamine (27 vs. 131 μM) and, most importantly, does not exhibit significant affinity for any other binding site [5]. That R-ketamine is agonistic may be inferred from a drug-discrimination study, where the recognition of ketamine was blocked by a σ1-antagonist [46]. These σ1 receptors are ligand-regulated transmembrane proteins expressed by neurons, microglia, astrocytes, and oligodendrocytes [47,48]. Agonists (compounds that mimic the effect of receptor overexpression) shift microglia-polarization from pro-inflammatory to reparative [47,49]. The σ1 receptor regulates Ca^2+^ entry through L-type Ca-channels [50], which activates the kinases CaMKII and CaMKIV, leading to Ras/MEK-ERK activation, Ser^133^P-CREB formation, and MEF2-mediated gene transcription [51,52]. Additionally, Ser^133^P-CREB enhances BDNF transcription [53,54]. Thus, the findings reported by Yao and colleagues [42] perfectly align with the notion that R-ketamine activates σ1 receptors. Various σ1-agonists have been tested for antidepressant activity in the forced swim test, the tail suspension test or the glucose-preference test, and all have demonstrated efficacy [55]. Therefore, σ1 receptor stimulation is considered a valid target for novel antidepressants [55,56], despite that two leading compounds, igmesine and cutamesine, have failed to reach the market [52]. The accumulated preclinical data on R-ketamine strongly support its potential use in MDD patients. Surprisingly and contrary to data from an early pilot study [57], a recent phase II trial found that R-ketamine in comparison to placebo, did not exhibit significant antidepressant efficacy at the 24 h time point (cited in [23]).

## 4. Is this the End of the R-Ketamine Development?

R-ketamine was administered intravenously at a dose of 0.5 mg/kg [57], which is comparable to the amount of R-ketamine present in racemic ketamine (usually dosed at 0.5–1 mg/kg; [58]). The affinity of S-ketamine for NMDA and μ-opioid receptors is 0.8 μM and 7 μM, respectively, whereas the affinity of R-ketamine for the σ1 site is relatively low (27 μM only) [5]. Therefore, it is possible that R-ketamine needs to be administered at higher doses in order to elicit a relevant σ1-mediated antidepressant response.

Another factor that should be considered is the timing of the measurement of the antidepressant response. The data summarized above suggest that S-ketamine causes NMDA blockade and μ-opioid receptor stimulation, leading to a fast-onset antidepressant effect, while R-ketamine stimulates σ1 receptors, which results in a slower-developing anti-inflammatory effect and a protracted antidepressant response. It is therefore possible that the 24 h time point might be too early to detect a robust antidepressant effect. Additionally, it is possible that the recruited patients had relatively low levels of inflammatory cytokines, which could impact the observed outcome.

Further possibilities arise from pharmacological interactions between σ1 activation and the activation of AMPA or μ-opioid receptors, or the blockade of NMDA receptors. Although hypothetical, such interactions are conceivable. For example, σ1 receptors physically interact with μ-opioid receptors and modulate their response [59]. Racemic ketamine has been shown to prevent and reverse μ-opioid receptor desensitization through a non-NMDA, ERK-dependent mechanism of action [60], although it remains to be determined if such an effect involves R-ketamine’s σ1-activity. Apart from μ-opioid receptors, interactions with other potentially relevant receptors have also been reported. The σ1 protein interacts with the NR1 subunit of the NMDA channel and with the BDNF receptor TrkB (summarized in [55,56]), but the functional consequences of these interactions have not yet been studied in detail. In this context, it is worth noting that BDNF-induced dendrite growth is stimulated by σ1 agonists [56] as well as by R-ketamine [42], though it remains to be determined if this is mediated via TrkB potentiation. 

## 5. The Antidepressant Pathways Activated by Racemic Ketamine and the Individual Enantiomers Converge on the Inhibition of GSK3β in Microglia

Microglia cells express μ-opioid receptors and TrkB [36,61]. Stimulation of μ-opioid receptors on microglia cells provokes PKC-mediated GSK3β inhibition and an anti-inflammatory response involving the suppression of the release of pro-inflammatory cytokines [41,62,63]. Paracrine BDNF release (from neurons, astrocytes, and endothelial cells) and autocrine microglial BDNF release activates TrkB, which triggers the PI3K-Akt signaling pathway, resulting in inhibitory phosphorylation of GSK3β [14,64,65]. So, the S-enantiomer of ketamine potentially inhibits microglial GSK3β by virtue of direct μ-opioid receptor stimulation, as well as indirectly by an increase in BDNF-TrkB-Akt signaling. The R-enantiomer of ketamine, too, may provoke GSK3β inhibition, since the σ1-induced MEK-ERK pathway results in activation of the AGC-kinase RSK, which also promotes inhibitory phosphorylation of GSK3β [66,67]. Finally, racemic ketamine owing to its high lipophilicity may give rise to physicochemical displacement of Gs from lipid rafts, which is thought to enhance the interaction with adenylate cyclase and to cause a rise in intracellular levels of cAMP [68]. The elevated levels of intracellular cAMP levels are thought to activate PKA, which like other AGC-kinases would produce the inhibitory phosphorylation of GSK3β [66]. Inhibition of microglial GSK3β activity is a common effect of numerous antidepressant compounds and principles [41]. Thus, racemic ketamine may well trigger four biochemical pathways involving the AGC-kinases, PKA, Akt, PKC, and RSK that ultimately all lead to GSK3β-inhibition. This could explain its unique therapeutic response rate in MDD.

## 6. Final Comments

The entire potential of the σ1 stimulant effect of R-ketamine remains to be fully uncovered. However, once established, it may boost the development of other σ1-agonists for major depression. The robust and rapid onset of ketamine’s antidepressant activity is believed to stem from its fortuitous combination of multiple antidepressant mechanisms. Given the positive interaction of σ1 receptors with both μ-receptors and BDNF-TrkB function [56], there is a good chance that the combination of S-ketamine and R-ketamine will give rise to a supra-additive antidepressant response (see Figure 1). Interestingly, the anti-suicidal activity of ketamine did not extend to S-ketamine [9]. The anti-suicidal effect of racemic ketamine was reported to be independent of NMDA-blockade mediated AMPA-mTOR signaling [69], which is in disagreement with the signaling pathway described for S-ketamine, but consistent with the responses triggered by R-ketamine. If R-ketamine is indeed responsible for therapeutic efficacy against suicidality, then it is likely that this effect will be shared by other σ1-agonists.

## Figures and Tables

**Figure 1 biomedicines-11-02664-f001:**
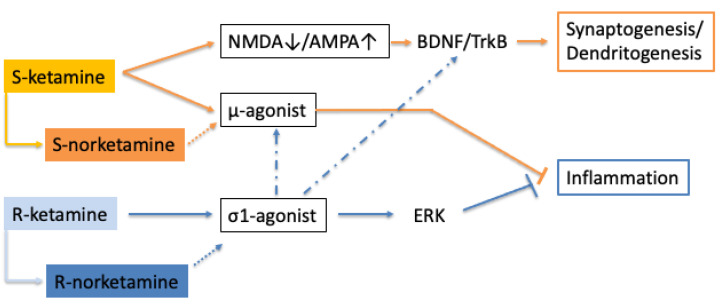
S-ketamine generates antidepressant activity via NMDA blockade and μ-opioid receptor activation, which, respectively, lead to synaptogenesis and anti-inflammatory efficacy. R-ketamine stimulates σ1 receptors, which also causes an anti-inflammatory and antidepressant response. S-norketamine and R-norketamine are primary metabolites. Their contribution to the pharmacological profile remains to be determined (dotted arrow). There is limited information that σ1 agonists may potentiate μ-opioid receptor and TrkB-mediated responses (indicated by a dash-dotted arrow).

## Data Availability

Not applicable.

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
