# Peer review of "Activation of σ1-Receptors by R-Ketamine May Enhance the Antidepressant Effect of S-Ketamine"

_biomedicines, 2023, doi:10.3390/biomedicines11102664_

Round 1
Reviewer 1 Report
The ms. “The role of σ1-receptor activation in the enigmatic antidepressant effect of ketamine” is an interesting commentary putting forward a hypothesis regarding different antidepressant properties between racemic ketamine and S-ketamine. Although this hypothesis needs to be experimentally proven, it is interesting and novel.
I would suggest only minor revisions (see below) and a language restyling.
- I would suggest to reformulate the title specifically mentioning the enantiomers and to remove the term “enigmatic”.
- The action of both S- and R-ketamine on AMPA receptors is not mentioned in either the abstract or the introduction.
- Lines 9-13: “The S-enantiomer acts as an NMDA-channel 9 blocker and as opioid μ-receptor agonist… that ultimately lead to inhibitory phos- 12 phorylation of GSK3β in microglia”; the section is extremely simplistic, uncomplete, and unclear.
- Lines 20-22: “In the current commentary it is argued that R-ketamine, due to its σ1- 20 activity not only provides an additional antidepressant mechanism, but in addition potentially enhances both the μ-receptor activity and the BDNF-TrkB signaling induced by S-ketamine”. Unclear sentence.
- Line 122 and following: the huge effort of Dr. Hashimoto to describe the mechanism of action of R-ketamine should be mentioned and discussed here. I understand that sigma 1 receptors are central in the view of the author, but other mechanisms can’t be ignored.
The quality is generally good but some sentences require language optimisation.
Author Response
I would like to thank the referee for his review, and for the time he has spent on doing so. Here are the responses to his suggestions.
1) Remove "enigmatic" from title. Simply leaving out the word “enigmatic” generates a scientifically incorrect sentence, because the sigma1 affinity is only true for the R-enantiomer. One possible solution would be: Activation of σ1-receptors by R-ketamine may enhance the antidepressant effect of S-ketamine. I have changed the title accordingly.
2) The action of both S- and R-ketamine on AMPA receptors is not mentioned in either the abstract or the introduction. I have added this to the introduction section (lines 37 and further, in red colour).
3) The S-enantiomer acts as an NMDA-channel .....(section is extremely simplistic, uncomplete, and unclear). These are the three primary pharmacological effects, which trigger an array of extracellular and intracellular effects, amongst which GSK3beta blockade is an essential point of convergency. Please note that this is in the summary section, where the word count is restricted to approximately 200 words. I see unfortunately no reasonable possibility for extension and clarification.
4) Unclear sentence line 20-22. To avoid confusion, I have removed the sentence about the lack of an early clinical effect of R-ketamine. I have reformulated the unclear sentence to: “The σ1-protein interacts with μ-opioid and TrkB-receptors, whereas in preclinical experiments σ1-agonists reduce μ-receptor desensitization and improve TrkB signal transduction. TrkB activation occurs as response to NMDA blockade. So, the σ1-activity of R-ketamine may not only enhance two pathways via which S-ketamine produces an antidepressant response, it furthermore provides an antidepressant activity in its own right.” With this effort, I am able to limit the word count to just under 200.
5) Dr Hashimoto's effort. To be honest, Professor Kenji Hashimoto is one of my ‘scientific heroes’. It is extremely admirable what he has discovered during the last decade. It should be noted that in my commentary I have cited seven of his most influential papers about the MoA of R-ketamine. In an earlier draft of the manuscript, I had a section about his discovery of the role of the hydroxy-metabolite of R-norketamine (2R,6R-HNK). But since it known that this metabolite acts as mGluR2 antagonist (Zanos et al, PNAS, 2019), and thus in principle acts as an indirect AMPA agonist, it would not have added additional weight to the main message of the manuscript (but instead add an unnecessary complication and dilution of the main message).
Reviewer 2 Report
This very interesting manuscript is devoted to investigation a role role of sigma receptors not only in antidepressant , but also the anti-suicidal action of ketamine with indicating molecular mechanisms. It is not entirely clear the data about absence antidepressant effect R-ketamine in the early stages clinical trial at the 24-hour time point (cited in reference 23). Usually, 10-14 days of antidepressant use is necessary for the development of antidepressant efficacy, given the activation of the BDNF synthesis and its accumulation. I think that in the final discussion, more possible use of ketamine as a sigma receptor agonist in clinical practice can be given.
Author Response
I would like to thank the referee for the time he has devoted in order to review my manuscript. The referee is entirely correct that R-ketamine should be given a second chance. At the 24 h time point, I expect that BDNF transcription and translation have occurred, which then can contribute to the resolution of the inflammation and the normalization of the microglia hyperactivity. This will take more time than just 24 hours. I expect that R-ketamine, owing to its sigma1 activity, will be a useful contribution to the therapeutic arsenal, but first this needs to be proven. Before that, it is probably not really helpful to write big speculative paragraphs.